# Zoledronate Causes a Systemic Shift of Macrophage Polarization towards M1 In Vivo

**DOI:** 10.3390/ijms22031323

**Published:** 2021-01-28

**Authors:** Manuel Weber, Andi Homm, Stefan Müller, Silke Frey, Kerstin Amann, Jutta Ries, Carol Geppert, Raimund Preidl, Tobias Möst, Peer W. Kämmerer, Marco Kesting, Falk Wehrhan

**Affiliations:** 1Department of Oral and Maxillofacial Surgery, Friedrich-Alexander University Erlangen-Nürnberg (FAU), 91054 Erlangen, Germany; andi.homm@gmail.com (A.H.); stefanchrismueller@googlemail.com (S.M.); jutta.ries@uk-erlangen.de (J.R.); raimund.preidl@uk-erlangen.de (R.P.); tobias.moest@uk-erlangen.de (T.M.); marco.kesting@uk-erlangen.de (M.K.); Falk.Wehrhan@outlook.de (F.W.); 2Department of Internal Medicine 3–Rheumatology and Immunology, Universitätsklinikum Erlangen, Friedrich-Alexander University Erlangen-Nürnberg (FAU), 91054 Erlangen, Germany; silke.frey@uk-erlangen.de; 3Department of Nephropathology, Institute of Pathology, Friedrich-Alexander University Erlangen-Nürnberg (FAU), 91054 Erlangen, Germany; kerstin.amann@uk-erlangen.de; 4Institute of Pathology, Friedrich-Alexander University Erlangen-Nürnberg (FAU), 91054 Erlangen, Germany; carol.geppert@uk-erlangen.de; 5Department of Oral and Maxillofacial Surgery, University of Mainz, 55122 Mainz, Germany; Peer.Kaemmerer@unimedizin-mainz.de

**Keywords:** bisphosphonate, Zoledronate, antiresorptives, macrophage polarization, M1, M2, immune modulation

## Abstract

Background: Immunomodulatory properties of bisphosphonates (BP) are suggested to contribute to the development of medication-associated osteonecrosis of the jaw (MRONJ). Furthermore, bisphosphonate-derived immune modulation might contribute to the anti-metastatic effect observed in breast cancer patients. Macrophages are potential candidates for the mediation of immunomodulatory effects of bisphosphonates. The study aimed to investigate the influence of bisphosphonates alone and in combination with surgical trauma on systemic macrophage polarization (M1 vs. M2) using an in vivo rat model. Methods: A total of 120 animals were divided into four groups. Groups 2 and 4 were treated with 8 × 40 μg/kg body weight of the BP Zoledronate i.p. (week 0–7). Groups 3 and 4 were exposed to surgical trauma (week 8, tooth extraction + tibia fracture), whereas in Group 1 neither medication nor surgical trauma was applied. After 8, 10, 12 and 16 weeks, skin, lung and spleen were immunohistochemically examined for macrophage polarization via expression analysis of CD68, CD163 and iNOS using a tissue microarray (TMA). Results: A significant shift of macrophage polarization towards M1 was observed in skin, spleen and lung tissue of animals, with and without surgical trauma, treated with BP when compared to those without BP application. Surgical trauma did not cause a significant increase towards M1 polarization. Conclusions: BP application leads to a systemic pro-inflammatory situation in vivo, independent of surgical trauma, as evidenced by the shift in macrophage polarization towards M1 in various somatic tissues. This provides a possible explanation for the clinically observed anti-tumor effect of bisphosphonates and might also contribute to pathogenesis of MRONJ.

## 1. Introduction

Bisphosphonates (BP) such as Zoledronate are commonly prescribed antiresorptive drugs. They inhibit osteoclast function and thus prevent bone resorption [1,2,3]. Therefore, they are used to treat osteoporosis and bone metastases. However, recent data indicate an anti-tumoral effect of BP beyond growth inhibition of bone metastases [4,5]. In brief, BP therapy provoked a decreased risk of developing bone metastases in early stage breast cancer patients [5]. Additionally, in the subgroup of postmenopausal women, a significantly increased survival in the BP-treated group was shown [5]. This effect cannot be explained by BP-mediated reduced bone resorption alone [4].

BP-medication related osteonecrosis of the jaw (MRONJ) (BP) is one of the most severe complications of BP treatment. MRONJ is characterized by necrotic jaw bone that is exposed to the oral cavity due to a lack of epithelial coverage. It can lead to loss of teeth, infections, jaw fractures and eventually to the loss of large parts of the jaw, requiring extensive surgical reconstruction [1,6]. The pathogenesis of MRONJ is not yet completely understood but it is suggested that BP reduces bone metabolism and turnover—especially in jaw bone [1]. The different developmental biological origin of jaw bone and extracranial bone is believed to be relevant for the jaw specificity of the disease [7,8]. In addition, immunologic parameters seem to contribute to the pathogenesis of MRONJ. In this regard, macrophages are supposed to be of particular importance [9].

The immunological microenvironment of the affected tissues and, in this context, the polarization of macrophages (M1 vs. M2) is involved in the pathogenesis of inflammatory and malignant diseases. M1-polarized macrophages act as pro-inflammatory and are associated with tissue destruction but also with tumor defense [10,11]. In contrast, M2 macrophages are immune-regulatory, contribute to wound healing and tissue repair and promote tumor growth by immunosuppressive features [12,13]. CD68 is a commonly used pan-macrophage marker in rats [14]. CD163 is frequently used for staining M2 macrophage in rat specimens, while iNOS is an established M1 marker [15,16].

In human jaw bone samples affected by BRONJ (BP), a significant increase of macrophage infiltration as well as M1 macrophage polarization was demonstrated [9]. These data are supported by in vitro experiments: The BP Zoledronate increased M1 polarization in TEP-1 macrophage-like cells [17]. In mice, an increased expression of TLR4 and increased M1 polarization of macrophages were detected after Zoledronate administration in vitro and in vivo [18].

A potential systemic shift of macrophage polarization in BP-treated individuals might explain the anti-metastatic effect of BP as well as the occurrence of MRONJ. It is well-accepted that the cell density and polarization of macrophages contribute to the progression of several malignancies [19]. In this regard, M2 polarized macrophages are considered to be tumor-promoting [10,11]. In accordance, in breast [20], lung [21] and oral cancer [22] an association of increased infiltration with CD163-positive M2 macrophages with inferior survival was shown. In oral cancer, tumor-infiltrating M2 macrophages were associated with occurrence of lymph node metastases [23]. Additionally, recent data revealed a link between M2 macrophage polarization and the initiation of oral cancer caused by initiated immunosuppression [24]. On the other hand, a shift towards M2 polarized macrophages is essential for the later stages of wound healing to promote tissue regeneration and wound closure [25,26].

The effects of BP treatment can be examined not only by studying the place of surgery and at the locally induced defect. The spleen is a secondary immune organ. It is involved in the regulation of immune responses locally and in the whole body. Additionally, it is of importance for inflammatory and degenerative diseases and modulates the innate and adaptive immunity. Spleen, lung and skin are populated with cells of the macrophage system. Therefore, BP treatment and/or a distant surgical trauma can also lead to a shift towards pro-inflammatory M1 polarization of macrophages in these organs.

In a Wistar rat model, the effect of the bisphosphonate Zoledronate on infiltration and polarization (M1 vs. M2) of tissue macrophages in different organs (spleen, lung and skin) was investigated in this study. It was tested if Zoledronate caused a systemic shift of macrophage polarization towards M1. Additionally, the contribution of surgical trauma to bisphosphonate-mediated alterations in macrophage infiltration and polarization was analyzed.

## 2. Results

For this analysis, the samples of 108 animals were available. The drop-outs were caused by animals that had to be sacrificed because abort criteria (anesthesiologic complications during surgery, post-operative infection) occurred prior to tissue harvesting.

### 2.1. General Considerations

The macrophage markers analyzed in this study showed a staining of the plasma membrane and the cytoplasm that was analogue to previously published macrophage analyses in human cancer and bone specimens [9,27]. Exemplary micrographs of the macrophage staining patterns are given in Figures 5a–c and 6. There was no relevant correlation between the time of sacrifice and the measured macrophage infiltration and polarization. Therefore, the time points 8, 10, 12 and 16 weeks after the start of the experiment were summarized for all analyses.

### 2.2. Macrophage Infiltration and Polarization in the Skin in Relation to Bisphosphonate Application

Considering all cases (surgical trauma and no surgical trauma), the CD68 labeling index in the subepithelium of the skin tissues (dermis and subcutaneous tissue) in bisphosphonate (BP)-treated animals was significantly increased compared to untreated controls (median 6.66% and 3.47%, respectively; *p* = 0.004; Table 1). Additionally, the iNOS labeling index was significantly higher in BP-treated rats compared to untreated animals (median 2.11% and 1.18%, respectively) (*p* = 0.017) (Table 1). In contrast, there was no difference in the relative cell density of CD163 positive M2 macrophages (Table 1). A similar pattern was seen analyzing cases without surgical trauma only. Animals without surgery showed a significantly increased CD68 labeling index in the subepithelial layer of the skin compared to those with no BP application (median 6.72% and 3.96%, respectively; *p* = 0.014; Table 1, Figure 1a). The iNOS labeling index in the skin subepithelium was significantly higher in BP-treated animals without surgical trauma compared to untreated controls (median 2.99% and 1.12%, respectively) (*p* < 0.001) (Table 1, Figure 1b). There was no significant difference with regard to CD163 expression (Table 1, Figure 1c). Considering only cases with surgical trauma, there were no significant differences in the expression of macrophage markers with regard to the application of bisphosphonates (Table 1).

The expression ratios between the different macrophage markers can be interpreted as indicators of macrophage polarization [9,27]. Considering all cases (surgical trauma and no surgical trauma), the CD163/CD68 ratio (indicator of M2 polarization) in the skin subepithelium was significantly lower in BP-treated rats compared to untreated animals (median value 2.24 and 3.38, respectively) (*p* = 0.007) (Table 1). Accordingly, the ratio between iNOS expressing cells and CD163 expressing cells (iNOS/CD163 ratio; indicator of M1 polarization) in BP treated animals was significantly higher (median value 0.18) than in those without BP application (median value 0.11) (*p* = 0.012) (Table 1). Considering only cases without surgical trauma, there was no significant difference in the iNOS/CD68 and the CD163/CD68 expression ratio (Table 1, Figure 1d,e); however, the iNOS/CD163 ratio (indicator of M1 polarization) in the subepithelial skin layer of BP-treated animals was significantly higher compared to untreated controls (median value 0.22 and 0.09, respectively) (*p* < 0.001) (Table 1, Figure 1f). Expression values for cases with surgical trauma are summarized in Table 1.

In the epithelial compartment of the skin, low levels of macrophage infiltration were detected. The epithelial macrophage expression is provided in the Appendix A.

### 2.3. Macrophage Infiltration and Polarization in the Lung in Relation to Bisphosphonate Application

In lung specimens, a significantly increased iNOS labeling index was seen in animals receiving bisphosphonate (BP) application compared to controls without antiresorptive medication (Table 2). This relation was observed in rats with surgical trauma, without surgery and in combined groups (Table 2). Considering the cases with no surgical trauma, the iNOS labeling index in lung tissue in BP-treated rats was 1.68% compared to 0.80% in untreated controls (*p* = 0.001) (Table 2, Figure 2b). In contrast, the CD68 and the CD163 labeling index in lung tissue was not significantly different between BP-treated and untreated animals (Table 2, Figure 2a,c).

Considering cases without surgical trauma, the iNOS/CD68 ratio (indicator of M1 polarization of macrophages) in lung tissue was significantly higher in BP-treated animals compared to untreated controls (median value 0.54 and 0.24, respectively) (*p* = 0.005) (Table 2, Figure 2d). Accordingly, the CD163/CD68 ratio (indicator of M2 polarization) in the lung was significantly lower in animals with BP application compared to untreated controls (median value 0.18 and 0.24, respectively) (*p* = 0.027) (Table 2, Figure 2e). The iNOS/CD163 expression ratio (indicator of M1 polarization) in lung tissue was significantly higher in BP-treated animals compared to untreated controls (median value 3.35 and 1.02, respectively) (*p* < 0.001) (Table 2, Figure 2e). Similar expression ratios were detected in animals with surgical trauma or in combined groups. These data are depicted in Table 2.

### 2.4. Macrophage Infiltration and Polarization in the Spleen in Relation to Bisphosphonate Application

As expected, the red pulp of spleen specimens showed the highest expression of macrophage markers. In the red pulp, a significantly decreased infiltration of CD163 positive cells was observed in cases with bisphosphonate (BP) application compared to untreated animals of all subgroups (cases with surgery, cases without surgery and all cases) (Table 3). In cases without surgical trauma, the CD163 labeling index was 34.17% in rats with BP treatment compared to 47.58% in untreated animals (*p* = 0.006) (Table 3, Figure 3c). In contrast, there was no significant difference regarding the CD68 and iNOS labeling index in spleen red pulp tissue between BP-treated and untreated animals (Table 3, Figure 3a,b).

Considering groups without surgical trauma, there was a significant increase in the iNOS/CD68 expression ratio (M1) in BP-treated animals compared to rats with no treatment (median value 2.73 and 2.37, respectively) (*p* = 0.03) (Table 3, Figure 3d). Similar expression ratios were seen considering all cases (*p* = 0.008) or only cases with surgical trauma (*p* = 0.166) (Table 3). The CD163/CD68 expression ratio (M2) was significantly lower in BP-treated animals compared to controls. Without surgery, the CD163/CD68 ratio was 1.29 in rats with BP treatment compared to 1.68 in untreated rats (*p* = 0.02) (Table 3, Figure 3e). Accordingly, the iNOS/CD163 ratio (M1) in spleen red pulp tissue in BP- treated rats was significantly increased (Table 3). In the group without surgery, the iNOS/CD163 ratio was 2.03 in BP-treated animals compared to 1.48 in untreated (*p* = 0.002) (Table 3, Figure 3f).

The white pulp of the spleen showed a significantly lower infiltration of macrophages compared to the red pulp (Figure 5a–c). Low iNOS and CD68 along with an extremely low CD163 expression were observed in the white pulp compared to the red pulp (Figure 5, Appendix A). In BP-treated animals, the CD163 labeling index in the spleen white pulp was significantly lower compared to untreated controls (Appendix A).

### 2.5. Macrophage Infiltration and Polarization in the Skin Subepithelium in Relation to Surgical Trauma

There was no significant difference in macrophage infiltration and macrophage polarization in the subepithelial layer of the skin with regard to the presence or absence of surgical trauma in rats not treated with bisphosphonate (BP). However, considering only BP-treated animals, a significantly decreased iNOS labeling index as well as iNOS/C68 and iNOS/CD163 expression ratio in rats with surgical trauma compared to rats without surgery was seen (see Table 1).

### 2.6. Macrophage Infiltration and Polarization in the Lung in Relation to Surgical Trauma

In lung tissue of rats with surgical trauma, a significantly increased iNOS labeling index was detected compared to animals without surgical intervention (Table 2). This relation was seen in animals with BP treatment, animals without antiresorptives and in both groups combined (Table 2). Considering the cases with BP application, the iNOS labeling index in lung tissue in surgically treated rats was 1.59% compared to 0.80% in controls without surgery (*p* = 0.001) (Table 2, Figure 2b). The highest iNOS labeling index was seen in BP treated rats that received surgery (median labeling index 3.15%) (Table 2). In contrast, CD68 and CD163 in lung tissue were not significantly different between rats with and without surgical intervention (Table 3). Additionally, the iNOS/CD68 ratio and the iNOS/CD163 ratio in cases with surgical trauma were higher compared to cases without surgical trauma (Table 2). The difference was significant when cases without surgical trauma or all cases combined were considered (Table 2). The highest expression ratios were reached in rats with surgical trauma and BP application. However, there was no significant difference between rats with and without surgery if only animals with BP treatment were considered (Table 2).

### 2.7. Macrophage Infiltration and Polarization in the Spleen Red Pulp in Relation to Surgical Trauma

In the red pulp of the spleen, an increase of macrophage infiltration in the group with surgical trauma was observed (Table 3). Considering groups with and without bisphosphonate (BP) application, the increase of CD68, iNOS and CD163 labeling index in animals with surgical trauma was significant (Table 3). In rats without BP application, a significant increase of the CD163 labeling index in animals with surgical trauma was evident (*p* = 0.031) (Table 3).

## 3. Discussion

### 3.1. Bisphosphonate-Derived Alteration of Systemic Macrophage Polarization and the Possible Influence on MRONJ

The current study revealed that application of the bisphosphonate (BP) Zoledronate leads to a systemic shift of the polarization of tissue macrophages towards M1. The M1-shift was observed in the skin, spleen and lung of the BP-treated Wistar rats when compared to untreated controls. The BP-induced M1 shift was noted in cases without and with surgical trauma. This could explain the anti-metastatic effect of BP and contribute to the pathogenesis of MRONJ.

In MRONJ, wound healing is impaired [6]. Macrophages are relevant in the healing process, especially in later stages. In mouse wound models, an increased infiltration of macrophages until day 2 post-injury and a decrease in steady state levels until day 14 was observed [25]. In vivo, macrophage polarization can be considered as a continuum between the extremes M1 and M2. M1 cells have high phagocytic activity and produce pro-inflammatory cytokines such as TNF alpha and IL-6 [25]. M1 macrophages dominate the early phase of wound healing and are responsible for removing pathogens and apoptotic neutrophils in the wound [25]. About 85% of macrophages are M1 polarized in early phase wounds. This ratio switches during wound maturation. In post-injury days 5–7, an M2 polarization dominated, with only about 15–20% of macrophages being M1 polarized [25].

It is believed that the initially infiltrating M1 macrophages repolarize towards M2 during healing processes [28]. The M2 macrophage-derived cytokines stimulate fibroblast and keratinocyte proliferation as well as neovascularization in the later phases of wound healing [25]. In chronic wounds, the switch towards M2 is lacking [25,29]. In MRONJ (BP)-affected human bone samples, an increased M1 and decreased M2 polarization of macrophages was seen [9]. In human and murine MRONJ (BP) mucosa samples, an increase of M1 polarization was also proven [30]. The severity of MRONJ could be reversed by an adoptive transfer of ex vivo expanded M2 macrophages in a mouse model [30]. The current study indicates that BP-derived changes in macrophage polarization might be systemic and might affect tissue macrophages in different organs. These data give additional evidence that BP-derived modulation of macrophages contributes to the pathogenesis of MRONJ.

### 3.2. Bisphosphonate-Derived Alteration of Systemic Macrophage Polarization and the Possible Influence on Cancer

The anti-metastatic effect of BP even in cases with an absence of bone metastases must be mediated by BP-derived action that goes beyond the inhibition of osteoclastic bone resorption [4]. Several other anti-tumoral actions of BP are discussed. BP may reduce the number of circulating tumor cells or the homing of these circulating cells to the bone [4]. In bone marrow aspirates of BP-treated breast cancer patients, a decreased count of disseminated tumor cells was seen [31]. Additionally, a direct toxic impact of BP on tumor cells is possible. This effect was apparent especially when bisphosphonates were combined with chemotherapy [4] or laser photochemotherapy [32]. Moreover, an immune modulating effect of BP that might be mediated via macrophages is discussed [4]. The current study gives evidence that BP has a long-term effect on tissue macrophages in vivo. The polarization of tissue macrophages shows a systemic switch from tumor-promoting M2 macrophages towards anti-tumoral and pro-inflammatory M1 macrophages. In the current experiment, this switch was not associated with the time interval after BP application.

In breast cancer, macrophages can contribute to up to 50% of total cell mass and are the main players in tumor immunology [33]. A connection between tumor progression and M2 polarized macrophages was proven [20,34]. Additionally, breast cancer cells were shown to mediate the shift of the macrophage population towards M2 in vitro causing an immunosuppressive tissue environment and tumor escape from the immune system [34]. A BP-derived shift of macrophage polarization towards M1 might therefore counteract tumor progression. An inhibition of CCL2 signaling was shown to decrease macrophage infiltration and showed anti-tumoral effects in breast cancer [35]. Zoledronate can suppress CCL2 signaling [19] and might therefore act as macrophage-modulating.

There are two main lineages of macrophages described: firstly, tissue-resident macrophages that originate from embryonic precursors and have the potential of self-renewal, and secondly, bone-marrow-derived macrophages that infiltrate from the blood stream [26]. The tissue-resident macrophages are derived from the yolk sac during embryogenesis [36]. Alveolar macrophages in the lung mainly consist of tissue-resident macrophages, while those in the skin are bone marrow derived [26]. Macrophages in the red pulp of the spleen originate from both the tissue-resident and the bone-marrow-derived lineage [36]. As the current study could show a M1 polarizing effect of BP on tissue macrophages in the skin, the lung and the spleen red pulp, the influence of BP on macrophage polarization seems to be independent of the lineage of tissue macrophages (tissue-resident vs. bone-marrow-derived). To prove this, additional analyses of lineage-defining analyses would be necessary.

It was shown that spleen red pulp macrophages are capable of inducing regulatory T-cells (T_reg_). This is mediated via the M2 cytokine TGF-beta [36]. T_reg_ cells act immunosuppressive and are considered as an important tumor-promoting cell type [37]. A reduction of the T_reg_ generation through reduced M2 polarization in the red pulp could also contribute to BP-derived anti-metastatic immune modulation.

The monoclonal RANKL antibody Denosumab is used as the first choice for the antiresorptive treatment of postmenopausal osteoporosis as an alternative to BP [38]. In myeloma patients, there was no inferiority of denosumab compared to Zoledronate with regard to the occurrence of skeletal-related events [39]. In most solid malignancies, Denosumab showed comparable or superior results regarding the incidence of bone complications [40]. However, considering the potential anti-tumoral immune-modulating effects of BP indicated by the current study and previous reports, comparisons between the anti-cancer effects of BP and Denosumab are needed. This is particularly important because Denosumab can also lead to osteonecrosis of the jaw. Moreover, comparing analyses of bisphosphonates and Denosumab on polarization and function would be desirable.

In contrast to previous in vitro studies that described a decreased viability of macrophages caused by BP [41,42], we did not see a decreased macrophage density in our current in vivo analysis. Although, there might be direct toxic effects of BP on macrophages in vivo, a shift of macrophage polarization seems to be one relevant factor.

### 3.3. Influence of Surgical Trauma on Systemic Macrophage Polarization

While the administration of bisphosphonates showed a significant M1 polarizing effect on macrophages in all investigated tissue types, the systemic effect of surgical trauma was less significant. In lung tissues, an increased iNOS labeling index in animals with surgical trauma was seen. This indicates a shift towards M1 polarization. However, the other macrophage markers showed no significant difference. In the red pulp of the spleen, an increase of all analyzed macrophage markers was seen. However, in most groups there was no significant change in the macrophage expression ratios. This indicates that surgical trauma does not have long lasting systemic effects on macrophage polarization. In contrast to this, a local modulation of macrophage polarization in cancer tissue caused by incision biopsies is shown [22,27].

### 3.4. Macrophages Polarization and Its Potential Influence on Dental Implant and Bone Grafting Success

Besides the effect of bisphosphonate application on macrophage polarization, the proportion of M1 and M2 polarized macrophages could be a possible marker for the description of graft incorporation and dental implant osseointegration in a physiological as well as in a compromised wound healing situation. In this context, the induction of an M1 polarization with a consequent pro-inflammatory tissue reaction could be a reason that the application of xenogeneic grafting materials did not show a sufficient osseous graft incorporation. However, a significant connective tissue formation within and adjacent to the graft could be observed [43,44]. Furthermore, M1 polarized macrophages could be responsible for the inferior grade of osseointegration of dental implants inserted in diabetic animals compared to dental implants in a physiological metabolism [45] or different dental implant surfaces influence the dynamics of peri-implant bone remodeling [45,46]. Based on the mentioned preclinical results, the evaluation of the M1 and M2 distribution in a population of macrophages in tissues could represent a potential biologic indicator for dental implant and bone grafting success in the long term.

### 3.5. Limitations of the Study

Currently, there is no clear evidence indicating which kind of dosing of BP in rat models best reflects the clinical situation in human patients. There are several different rat models described in the literature with regard to BP dosing regimens and treatment duration [47,48,49]. Compared to these models, we decided to choose a relatively low dosing of Zoledronate to avoid a direct toxic effect of the bisphosphonate.

The induced trauma in this study had no significant influence on systemic macrophage polarization. It may be that a more aggressive surgical trauma might have stronger effects on the expression of macrophage markers.

Another possible limitation is that there might be differences in macrophage polarization between rats and humans. However, the selected macrophage markers were successfully used in previous human studies and animal experiments.

The regions of the stained cells in the CD68 staining do not match that of iNOS (Figure 6). NO is an important pro-inflammatory mediator with an immense effect on the immune system and different immune cells. Thus, the missing correlation between CD68 and iNOS could be explained by the fact that iNOS is produced by a variety of cells after induction by cytokines or other stimuli. Indeed, many studies have shown that iNOS is expressed by cells other than macrophages such as activated T cells and mature dendritic cells (mDCs) [50]. Therefore, this discrepancy could be attributed to an iNOS expression in other immune cells. However, iNOS-expressing T-cells show immune activating effects similar to iNOS positive M1 macrophages [50]. An exact determination of each individual cell type in immunohistochemistry would require a multi-marker-staining, which should be done in further studies. Additionally, the authors hypothesize that surgery and BP treatment have a direct and primary impact on the tibia and jaw. This point will be addressed by a further study.

## 4. Materials and Methods

### 4.1. Animal Experiment

For this analysis, an animal experiment with 120 Wistar rats was performed. The experiment was part of the DFG project WE52731/1-1. Animal experiments were approved by the local authorities “Regierung von Mittelfranken” No 54-25321-3/09. The Wistar rats were divided into 4 groups. Group 1 (*n* = 30) included animals which were not untreated and served as control. For 8 weeks, the animals of Groups 2 (*n* = 32) and 4 (*n* = 20) were treated with the amino-bisphosphonate Zoledronate (Zometa, Novartis Pharma, Basel, Switzerland) at a dose of 40 μg/kg body weight i.p. one time per week (Table 1). The animals of Groups 3 (*n* = 26) and 4 (*n* = 20) were subjected to surgical treatment that consisted of extraction of the mandibular first molar on one side of the jaw and the osteotomy of a unilateral tibia bone followed by an immediate rigid osteosynthesis with an 8-hole 2.0 mini plate and 4 screws (Stryker, Kalamazoo, MI, USA) after 8 weeks (Table 4). Tooth extraction and tibia bone osteotomy were conducted under general anesthesia using 100 mg/kg of ketamine hydrochloride (Parke-Davis, Berlin, Germany) and 2.5 mg/kg of Xylacin (Bayer, Leverkusen, Germany) via intraperitoneal injection. Postoperative analgesia was done using 2 mg/kg i.p. Buprenorphin (Temgesic, Essex Pharma, Munich, Germany) for the first and second postoperative days and further if necessary. At 8, 10, 12 and 16 weeks, animals were sacrificed via overdose of anesthetics and bone (mandibular and tibia a bone) and different types of soft tissues (skin, lung, spleen) were harvested (Table 4). These tissues were formalin fixated and embedded into paraffin.

### 4.2. Design of Tissue Microarrays (TMA) 

Serial sections were prepared using a rotary microtome (Leica RM2165, Leica Biosystems, Wetzlar, Germany) at a thickness of 2 µm, and Hematoxylin and Eosin (H&E) staining was performed. All stained slides were scanned and digitized with the method of “whole slide imaging” in cooperation with the Institute of Pathology of the University of Erlangen-Nürnberg using a Pannoramic 250 Flash III Scanner (3DHistech, Budapest, Hungary). The digital slides were analyzed using the software Case Viewer (3DHistech). For establishing the tissue micro array, representative punch locations were annotated to each digital slide. For skin, the epithelia were placed in the middle and three punches of 1 mm diameter were taken from each sample (Table 5). For spleen, red and white pulp were marked and three punches of 1.5 mm diameter were obtained per sample. In the lung tissue, representative areas were marked and three punches of 1.5 mm diameter were assessed for each sample (Table 5; Figure 4). A total of 1035 punches were achieved, processed and analyzed. For the assembly of the TMA blocks, the device TMA Grand Master (3DHistech) was used. The digital HE slides were imported into the TMA Grand Master control software and aligned with the donor block images (Figure 4). Afterwards, the punching targets were confirmed and the machine created the TMA blocks automatically.

### 4.3. Immunohistochemical Staining and Quantitative Analysis

The staining procedure was performed using the LSAB2 K0609 kit (Dako Deutschland GmbH, Hamburg, Germany) and an autostainer (Dako Deutschland GmbH, Hamburg, Germany). The following primary antibodies were applied: anti-CD68 1:50 (MCA341R, Bio-Rad Laboratories, Hercules, CA, USA), anti-CD163 1:100 (MCA342R, Bio-Rad Laboratories, Hercules, CA, USA) and anti-iNOS 1:100 (Novocastra Leica, Nussloch, Germany). All TMA slides were scanned and digitized with the method of “whole slide imaging” in cooperation with the Institute of Pathology of the University of Erlangen-Nürnberg using a Pannoramic 250 Flash III Scanner (3D Histech, Budapest Hungary) as previously described [23,27].

All TMA slides were imported into the digital pathology image analysis software Tissue Studio (Definiens AG, Munich, Germany). This software analyzed the digital slides and detected automatically the TMA cores and the tissue (Figure 5). After training the software, it could independently divide the tissue into different regions of interests (ROI). The training was done by marking different ROI in the preview using machine learning technology.

For the different tissue types and the various stains, it was necessary to adjust the parameters according to the intensity of the stains. The parameter settings were identical for the tissue types and stains. Representative results are shown in Figure 6.

Assessment of spleen samples with anti-CD163 and anti-iNOS was performed using the cytoplasmatic analysis tool of Tissue Studio. The parameters hematoxylin threshold (0.07 and 0.08), marker threshold (0.35 and 0.3), typical nucleus size in µm^2^ (25 and 25) and the typical cell size in µm^2^ (450 and 350) were set. Because of the similarity of the spleen anti-CD68 staining to a nuclear marker staining, the nuclear marker analysis tool of Tissue Studio was used for counting of CD68-expressing cells. The parameters hematoxylin threshold (0.05), Immunhistochemical (IHC) threshold (0.55) and typical nucleus size in µm^2^ (52) were set. The spleen tissue was divided into these ROI: white pulp and red pulp.

For lung tissue with anti-CD68, anti-CD163 and anti-iNOS staining, the nuclear marker analysis tool of Tissue Studio was used and the parameters hematoxylin threshold (0.07; 0.07 and 0.06), IHC threshold (0.4; 0.4 and 0.35) and typical nucleus size in µm^2^ (35; 35 and 35) were set. For the skin tissue (CD68, anti and iNOS) the nuclear marker tool was used and the parameters hematoxylin threshold (0.07; 0.07 and 0.05), IHC threshold (0.4; 0.4 and 0.3) and typical nucleus size in µm^2^ (60; 60 and 60) were set. For analysis, the skin tissue was divided into epithelium (epidermis) and sub-epithelium compartment. The sub-epithelium contains the dermis and the subcutaneous tissue.

All parameters were set to ensure that the positive and negative cells shown in the Tissue Studio software preview corresponded approximately to human evaluation. Additionally, it was ensured that total cell count within a tissue type was approximately equal in different stainings for better comparability.

The Tissue Studio software calculated the number of positive and negative cells, and the tissue area per core and per ROI. For each tissue stain, the Tissue Studio software was run with the adjusted settings and the results were saved in a folder which contained a CSV file with all the positive and negative cells per core and ROI, and contained a CSV file with the area information (in mm^2^) for each core and ROI. Additionally, it contained screenshots of each core in original, with colored markings of positive and negative cells and with colored markings of the division into the different ROI.

### 4.4. Statistical Analysis

For statistical analysis, the high number of different result files, which were sorted by core, had to be rearranged. In the end, all necessary information was summed up in one Excel (Microsoft, Redmond, WA, USA) file. Therefore, a program in the programming language Python was written. Using the Python-based program, the cell and area information of all of the individual CSV result files from the Tissue Studio software were collected and data were summarized in one Excel table for the statistical analysis. Also, the ratios of the different stains (e.g., iNOS/CD68) were calculated and the data were added to the table.

To analyze the immunohistochemical staining, the cell count per mm^2^ was determined as the number of positive cells per mm^2^ within the tissue. The labeling index was calculated dividing the number of positive cells by the number of all cells (positive + negative). The data were investigated for a significant relationship between bisphosphonate application, surgical trauma and a redistribution of macrophage markers. The results were expressed as the median values. Also, standard deviation (SD) was determined. Box plot diagrams represent the median, the interquartile range, minimum (Min) and maximum (Max). The statistical analyses were performed with the software SPSS 22 for Mac OS (IBM Inc., New York, NY, USA) using the Mann-Whitney-U-Test. Two-sided, adjusted *p*-values ≤ 0.05 were considered to be significant.

## 5. Conclusions

To our knowledge, this is the first study performing a systematic analysis of the polarization of tissue macrophages in a bisphosphonate animal model. We can show a systemic shift of macrophage polarization towards M1 in bisphosphonate-treated animals. This was present in tissues with a dominating population of tissue-resident macrophages and in organs with predominantly bone-marrow-derived macrophages. The BP-derived shift from M2 towards M1 could contribute to the explanation of impaired wound-healing in BP-affected soft tissue and might also explain the clinically observed anti-metastatic effect of BP.

## Figures and Tables

**Figure 1 ijms-22-01323-f001:**
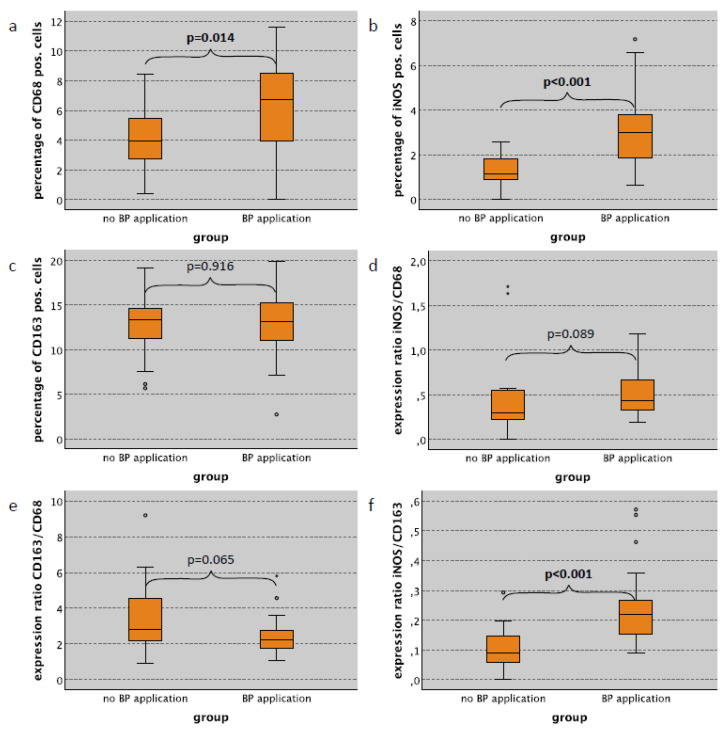
Influence of bisphosphonate application on macrophage infiltration and polarization in subepithelial skin tissue without surgical trauma. Expression of macrophage markers and the correlation to the bisphosphonate (BP) application in the subepithelial layer of the skin (skin subepithelium) of rats without surgical trauma. (**a**) Percentage of CD68 pos. cells depending on BP application (pos. cells/all cells in %). (**b**) Percentage of iNOS pos. cells depending on BP application (pos. cells/all cells in %). (**c**) Percentage of CD163 pos. cells depending on BP application (pos. cells/all cells in %). (**d**) Expression ratio of iNOS and CD68 depending on BP application ((iNOS pos./all cells)/(CD68 pos./all cells)). (**e**) Expression ratio of CD163 and CD68 depending on BP application ((CD163 pos./all cells)/(CD68 pos./all cells)). (**f**) Expression ratio of iNOS and CD163 depending on BP application ((iNOS pos./all cells)/(CD163 pos./all cells)). For statistical analysis, the Mann-Whitney-U-Test was used. Significant *p*-values (*p* < 0.05) are indicated in bold letters. * *p* < 0.05, ** *p* < 0.005.

**Figure 2 ijms-22-01323-f002:**
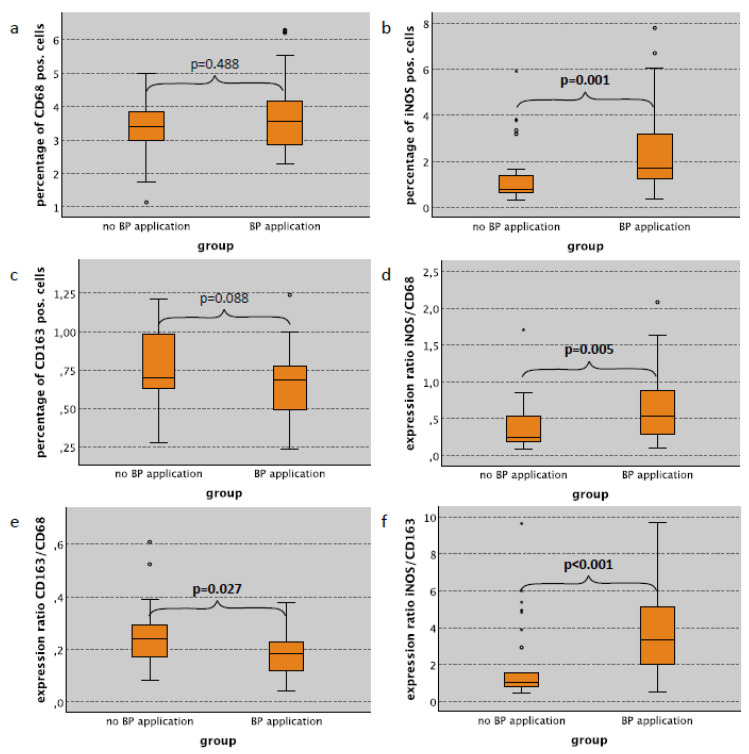
Influence of bisphosphonate application on macrophage infiltration and polarization in lung tissue without surgical trauma by analysis of the expression of different MP markers. (**a**) Percentage of CD68 pos. cells (pos. cells/all cells in %). (**b**) Percentage of iNOS pos. cells (pos. cells/all cells in %). (**c**) Percentage of CD163 pos. cells (pos. cells/all cells in %). (**d**) Expression ratio of iNOS and CD68 ((iNOS pos./all cells)/(CD68 pos./all cells)). (**e**) Expression ratio of CD163 and CD68 ((CD163 pos./all cells)/(CD68 pos./all cells)). (**f**) Expression ratio of iNOS and CD163 ((iNOS pos./all cells)/(CD163 pos./all cells)). Significant *p*-values (*p* < 0.05) are indicated in bold letters. For statistical analysis, the Mann-Whitney-U-Test was used. * *p* < 0.05, ** *p* < 0.005

**Figure 3 ijms-22-01323-f003:**
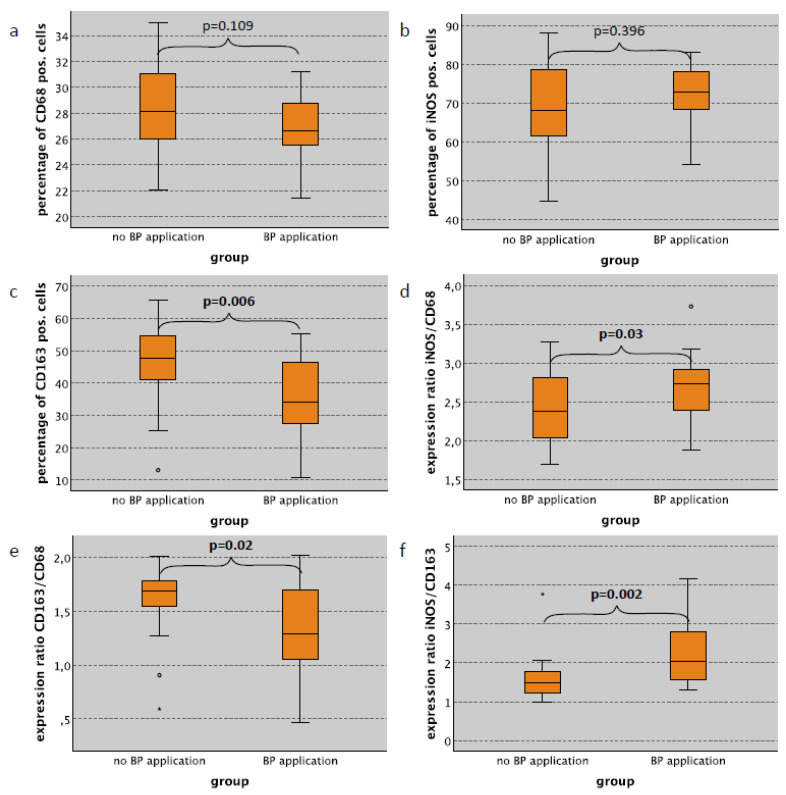
Influence of bisphosphonate application on macrophage infiltration and polarization in spleen red pulp tissue without surgical trauma. Expression of macrophage markers and the correlation to the BP application in the spleen red pulp of rats without surgical trauma. (**a**) Percentage of CD68 pos. cells (pos. cells/all cells in %). (**b**) Percentage of iNOS pos. cells (pos. cells/all cells in %). (**c**) Percentage of CD163 pos. cells (pos. cells/all cells in %). (**d**) Expression ratio of iNOS and CD68 ((iNOS pos./all cells)/(CD68 pos./all cells)). (**e**) Expression ratio of CD163 and CD68 ((CD163 pos./all cells)/(CD68 pos./all cells)). (**f**) Expression ratio of iNOS and CD163 ((iNOS pos./all cells)/(CD163 pos./all cells)). Significant *p*-values (*p* < 0.05) are indicated in bold letters. For statistical analysis, the Mann-Whitney-U-Test was used. * *p* < 0.05.

**Figure 4 ijms-22-01323-f004:**
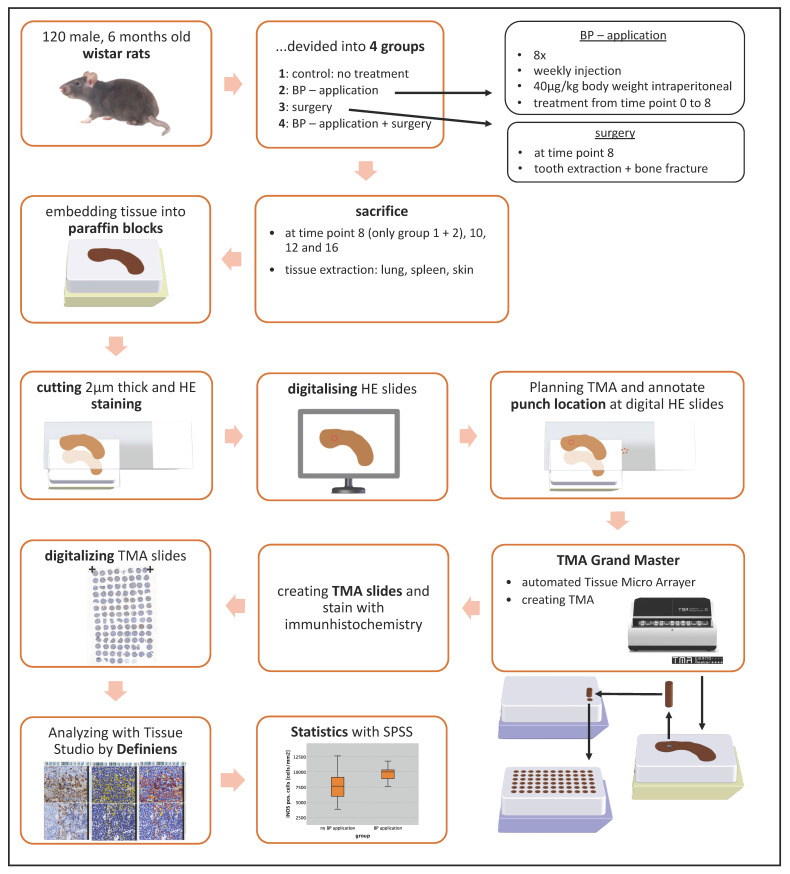
Experimental layout. Overview and visualization of the individual steps performed in this experiment. 3D sketches with kind permission of W. Müller.

**Figure 5 ijms-22-01323-f005:**
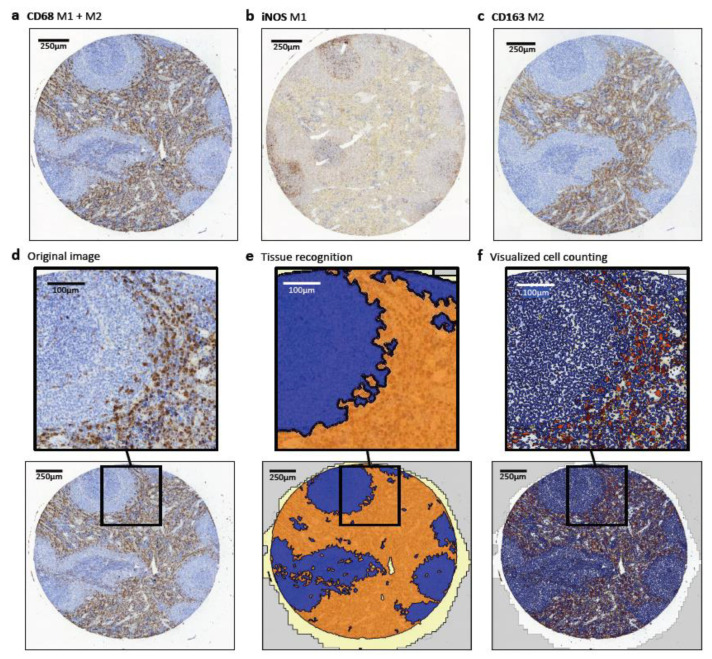
Staining and analysis of the TMA. Image analysis of Tissue Studio in a TMA core of spleen tissue. Expression of the macrophage markers CD68, CD163 and iNOS is shown. CD68 (**a**) is the best-established pan-macrophage marker for M1 and M2 polarized macrophages. iNOS (**b**) is used as a marker for M1 polarized macrophages. CD163 (**c**) is an established M2 macrophage marker. (**d**) Original image with CD68 staining of a TMA core used for image analysis with Tissue Studio. (**e**) Division into the different region of interests (ROI): white pulp (blue), red pulp (orange) and white space (light yellow) made with Tissue Studio. (**f**) Cell classification into CD68 negative cells (blue) and positive cells with the different intensities 1 (yellow), 2 (orange) and 3 (red).

**Figure 6 ijms-22-01323-f006:**
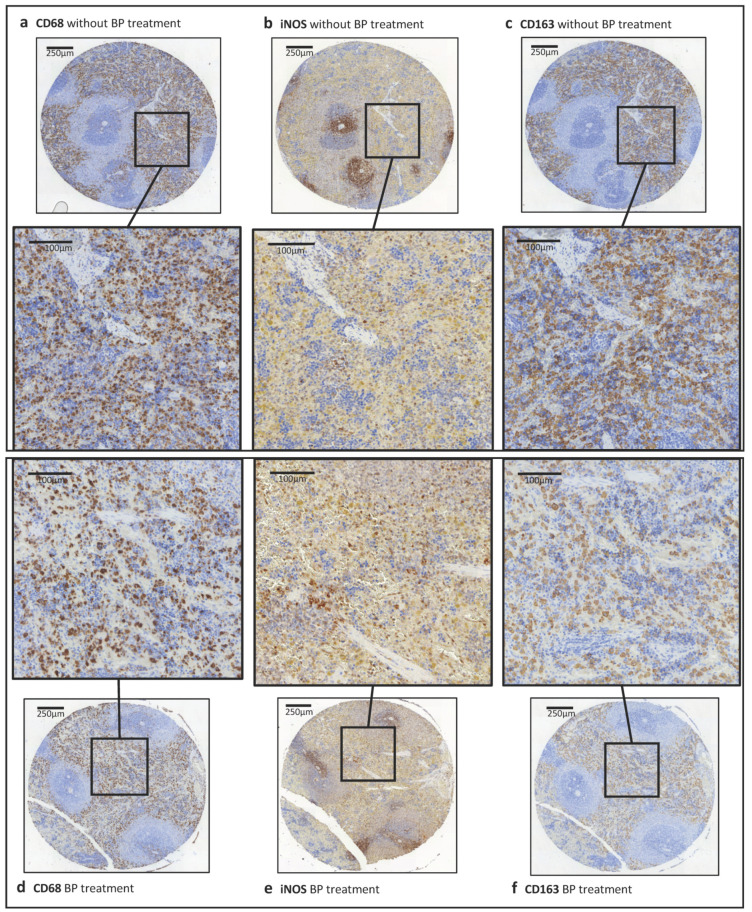
Comparison of the stainings from spleen tissue without and with BP treatment. Expression of the macrophage markers CD68, CD163 and iNOS distribution in a TMA core of spleen tissue of rats without (**a**–**c**) and with (**d**–**f**) BP treatment are shown, and their associated magnification.

**Table 1 ijms-22-01323-t001:** CD68. iNOS und CD163 expressions (pos. cells/all cells in %) and iNOS and CD163 to CD68 ratios. *p*-values were calculated by the Mann-Whitney-U-Test.

((pos. Cells/all Cells in %)/(pos. Cells/all Cells in %)) in Skin Subepithelium Tissue of Wistar Rats													
Marker	*n*	CD68		iNOS		CD163		Ratio iNOS/CD68		Ratio CD163/CD68		Ratio iNOS/CD163	
		Median	SD	Median	SD	Median	SD	Median	SD	Median	SD	Median	SD
**BP application**—all cases													
no	43	3.47	2.46	1.18	1.99	13.55	4.93	0.38	3.71	3.38	5.38	0.11	0.11
yes	33	6.66	3.17	2.11	1.75	13.20	5.10	0.36	1.31	2.24	2.58	0.18	0.14
*p*-value		**0.004**		**0.017**		0.892		0.906		**0.007**		**0.012**	
**BP application**—only no surgical trauma cases													
no	21	3.96	2.18	1.12	0.72	13.37	3.94	0.29	0.70	2.79	6.99	0.09	0.07
yes	23	6.72	3.05	2.99	1.74	13.20	3.75	0.44	0.67	2.22	2.11	0.22	0.13
*p*-value		**0.014**		**<0.001**		0.916		0.089		0.065		**<0.001**	
**BP application**—only surgical trauma cases													
no	22	3.42	2.74	1.31	2.63	13.97	5.81	0.42	5.14	3.61	3.22	0.13	0.14
yes	10	5.72	3.52	1.04	0.96	14.03	7.62	0.23	2.20	2.47	3.51	0.08	0.06
*p*-value		0.251		0.434		0.826		**0.047**		0.129		0.164	
**surgical trauma**—all cases													
no	44	5.21	2.84	1.87	1.61	13.29	3.80	0.40	0.68	2.45	5.23	0.15	0.13
yes	32	3.68	3.02	1.24	2.28	13.97	6.31	0.36	4.41	3.31	3.29	0.08	0.12
*p*-value		0.105		0.119		0.556		0.622		0.252		**0.03**	
**surgical trauma**—only no BP application cases													
no	21	3.96	2.18	1.12	0.72	13.37	3.94	0.29	0.70	2.79	6.99	0.09	0.07
yes	22	3.42	2.74	1.31	2.63	13.97	5.81	0.42	5.14	3.61	3.22	0.13	0.14
*p*-value		0.423		0.369		0.752		0.224		0.593		0.399	
**surgical trauma**—only BP application cases													
no	23	6.72	3.05	2.99	1.74	13.20	3.75	0.44	0.67	2.22	2.11	0.22	0.13
yes	10	5.72	3.52	1.04	0.96	14.03	7.62	0.23	2.20	2.47	3.51	0.08	0.06
*p*-value		0.428		**0.003**		0.63		**0.014**		0.857		**<0.001**	

**Table 2 ijms-22-01323-t002:** CD68. iNOS und CD163 expressions (pos. cells/all cells in %) and iNOS and CD163 to CD68 ratios. *p*-values were calculated by the Mann-Whitney-U-Test.

((pos. Cells/all Cells)/(pos. Cells/all Cells)) in Lung Tissue of Wistar Rats													
Marker	*n*	CD68		iNOS		CD163		Ratio iNOS/CD68		Ratio CD163/CD68		Ratio iNOS/CD163	
		Median	SD	Median	SD	Median	SD	Median	SD	Median	SD	Median	SD
**BP application**—all cases													
no	53	3.33	1.01	1.22	1.20	0.68	0.27	0.41	0.34	0.22	0.11	1.76	1.87
yes	52	3.63	2.09	2.31	2.31	0.69	0.27	0.71	0.66	0.18	0.09	3.81	3.11
*p*-value		0.209		**<0.001**		0.509		**0.002**		0.109		**<0.001**	
**BP application**—only no surgical trauma cases													
no	29	3.40	0.90	0.80	1.33	0.70	0.24	0.24	0.35	0.24	0.12	1.02	2.18
yes	32	3.54	1.35	1.68	1.78	0.69	0.22	0.54	0.46	0.18	0.08	3.35	2.41
*p*-value		0.488		**0.001**		0.088		**0.005**		**0.027**		**<0.001**	
**BP application**—only surgical trauma cases													
no	24	3.20	1.14	1.59	0.99	0.58	0.30	0.49	0.33	0.17	0.11	2.81	1.29
yes	20	3.67	2.89	3.15	2.80	0.69	0.32	0.79	0.86	0.18	0.10	4.20	3.92
*p*-value		0.258		**0.002**		0.239		**0.038**		0.832		**0.004**	
**surgical trauma**—all cases													
no	61	3.50	1.17	1.34	1.66	0.69	0.24	0.45	0.43	0.20	0.11	2.05	2.46
yes	44	3.58	2.16	2.00	2.23	0.66	0.31	0.53	0.66	0.18	0.10	3.51	3.00
*p*-value		0.59		**0.004**		0.608		**0.026**		0.413		**0.012**	
**surgical trauma**–only no BP application cases													
no	29	3.40	0.90	0.80	1.33	0.70	0.24	0.24	0.35	0.24	0.12	1.02	2.18
yes	24	3.20	1.14	1.59	0.99	0.58	0.30	0.49	0.33	0.17	0.11	2.81	1.29
*p*-value		0.872		**0.004**		0.054		**0.019**		0.116		**0.001**	
**surgical trauma**—only BP application cases													
no	32	3.54	1.35	1.68	1.78	0.69	0.22	0.54	0.46	0.18	0.08	3.35	2.41
yes	20	3.67	2.89	3.15	2.80	0.69	0.32	0.79	0.86	0.18	0.10	4.20	3.92
*p*-value		0.367		**0.024**		0.367		0.17		0.707		0.137	

**Table 3 ijms-22-01323-t003:** CD68. iNOS und CD163 expressions (pos. cells/all cells in %) and iNOS and CD163 to CD68 ratios. *p*-values were calculated by Mann-Whitney-U-Test.

((pos. Cells/all Cells in %)/(pos. Cells/all Cells in %)) in Spleen Red Pulp of Wistar Rats													
Marker	*n*	CD68		iNOS		CD163		Ratio iNOS/CD68		Ratio CD163/CD68		Ratio iNOS/CD163	
		Median	SD	Median	SD	Median	SD	Median	SD	Median	SD	Median	SD
**BP application**—all cases													
no	48	28.52	2.90	72.59	11.70	50.46	9.83	2.49	0.41	1.75	0.28	1.43	0.43
yes	44	27.51	2.38	75.64	7.16	39.93	11.04	2.72	0.33	1.37	0.39	1.88	0.91
*p*-value		0.116		0.099		**<0.001**		**0.008**		**<0.001**		**<0.001**	
**BP application**—only no surgical trauma cases													
no	26	28.15	3.10	68.18	12.15	47.58	11.91	2.37	0.44	1.68	0.31	1.48	0.55
yes	25	26.61	2.27	72.76	7.73	34.17	12.47	2.73	0.40	1.29	0.43	2.03	1.15
*p*-value		0.109		0.396		**0.006**		**0.03**		**0.02**		**0.002**	
**BP application**–only surgical trauma cases													
no	22	28.85	2.67	73.55	10.53	51.77	5.03	2.55	0.37	1.79	0.19	1.37	0.23
yes	19	28.68	2.09	76.43	4.59	42.69	7.59	2.70	0.23	1.48	0.31	1.82	0.33
*p*-value		0.676		0.075		**<0.001**		0.166		**<0.001**		**<0.001**	
**surgical trauma**–all cases													
no	51	27.19	2.79	72.53	10.28	43.36	13.18	2.64	0.44	1.63	0.40	1.63	0.96
yes	41	28.72	2.39	75.96	8.43	49.37	8.07	2.61	0.32	1.72	0.30	1.61	0.36
*p*-value		**0.013**		**0.011**		**0.011**		0.364		0.033		0.302	
**surgical trauma**—only no BP application cases													
no	26	28.15	3.10	68.18	12.15	47.58	11.91	2.37	0.44	1.68	0.31	1.48	0.55
yes	22	28.85	2.67	73.55	10.53	51.77	5.03	2.55	0.37	1.79	0.19	1.37	0.23
*p*-value		0.352		0.172		**0.031**		0.207		**0.012**		0.363	
**surgical trauma**—only BP application cases													
no	25	26.61	2.27	72.76	7.73	34.17	12.47	2.73	0.40	1.29	0.43	2.03	1.15
yes	19	28.68	2.09	76.43	4.59	42.69	7.59	2.70	0.23	1.48	0.31	1.82	0.33
*p*-value		**0.008**		**0.008**		0.07		0.878		0.231		0.562	

**Table 4 ijms-22-01323-t004:** Overview of the time schedule for animal treatment and tissue extraction.

Group	*n*	Total: 108 Animals, 6-Month-Old Male Wistar Rats				
1	30	no BP application	no surgery (control group)	sacrifice	sacrifice	sacrifice
		(control group)	sacrifice + tissue extraction	+ tissue extraction	+ tissue extraction	+ tissue extraction
2	32	40 μg/kg body weight Zoledronate	no surgery	sacrifice	sacrifice	sacrifice
		i.p. per week (8× at time point 0–7)	sacrifice + tissue extraction	+ tissue extraction	+ tissue extraction	+ tissue extraction
3	26	no bisphosphonate application	bone fracture + tooth extraction	sacrifice	sacrifice	sacrifice
				+ tissue extraction	+ tissue extraction	+ tissue extraction
4	20	40 μg/kg body weight Zoledronate	bone fracture + tooth extraction	sacrifice	sacrifice	sacrifice
		i.p. per week (8× at time point 0–7)		+ tissue extraction	+ tissue extraction	+ tissue extraction
time point		0	8	10	12	16
		start of experiment	weeks after start of experiment			

**Table 5 ijms-22-01323-t005:** Overview of the extracted soft tissue and the TMA punch targets.

Soft Tissue	Punch Target	Number of	Punch Diameter	Number of Punches	Number of
		Tissue Samples	in mm	Per Target	TMA Blocks
lung	lung tissue	103	1.5	3	3
spleen	red pulp	104	1.5	3	3
	white pulp				
skin	skin epithelium	102	1	3	5

## Data Availability

The data presented in this study are available on request from the corresponding author.

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
