# Peer review of "Zoledronate Causes a Systemic Shift of Macrophage Polarization towards M1 In Vivo"

_ijms, 2021, doi:10.3390/ijms22031323_

Round 1
Reviewer 1 Report
In this paper Weber et al performed a systematic analysis of the polarization of macrophages in bisphosphonate treated mice. I found this paper interesting and could potentially provide good new insight in the field. However, I believe it should be improved before it can be considered for publication.
Major:
In Figure 2b the staining with INOS is not well distinguishable and does not correlate with CD68, whereas CD163 seems to be expressed by nearly all macrophages in the section presented.
Authors should present a better example. Or explain why this discrepancy.
Authors should report representative stained sections of CD68, CD163 and INOS for WT and BP treated samples.
It is unclear to me if, why and how the surgery of tooth and Tibia should promote the M1 polarization of macrophages in lungs, and Spleen with or without bisphosphonate treatment. The authors should introduce the rational of this better.
Would help the reader to have a figure reporting all the groups surgery vs not surgery with and without BP at least for CD68+ and INOS+ with multiple comparison analysis together with the tables.
To this extend, why the authors considered only soft tissues? Are there any difference of macrophage infiltrate or M1 shift in the Jaw or tibia after surgery with or without bisphosphonate treatment?
In the figures presented, there is no actual increase in CD68+ macrophage numbers in the tissues both without surgery and with BP application.
What change it seems to be only the polarization. M1 proinflammatory macrophages are normally considered to derive from monocytes infiltrates in the tissue. These cells are known to express CCR2, whether tissue resident macrophages normally do not express CCR2. Could be interesting to see the CCR2 expression in the CD68+ INOS+ macrophages. It might answer the question whether the M1 shift induced by BP is due to infiltrate of monocytes or by some unknown effect of BP played directly on tissue resident macrophages.
Indeed, the author claim “As the current study could show a M1 polarizing effect of BP on t issue macrophages in the skin, the lung and the spleen red pulp, the influence of BP on macrophage polarization seems to be independent on the lineage of tissue macrophages (tissue-resident vs. bone-marrow-derived).” It is not fully supported by the data presented at this point.
Are there any differences among the groups in systemic cytokines like TNF and IL6 normally associated with M1 macrophages and proinflammatory state?
Minors:
The statistical analysis used should be indicated in every figure legend.
Would be visually more appealing to have different colors for Controls VS treated in all the figures.
Legends for the supplementary figures are missing. Please add.
From line 77 to 82, what does it mean “it should be examined if?” The last part of introduction should describe what the manuscript is presenting. These phrases should be rewritten.
Line 135 should be “detected”.
Several more typos are present in the manuscript, please consider proofreading the manuscript carefully.
Author Response
please see attchment.

Reviewer 2 Report
The authors investigated the role of zoledronate on systemic shift of macrophage polarization towards M1 in wistar rat model. Very interesting and would contribute to knowledge of possible pathogenesis of BP-associated complications.
'Surgical Trauma' is of concern. Although it doesn't show statistical significance, surgical trauma showed trend of lower CD68, iNOS values. Considering the surgical trauma is main risk factor for ONJ development and also deeply associated with initial inflammation and wound repair, this should not be overlooked. The reviewer recommends additional analysis with adjustment of surgical trauma also with consideration time point such as RM ANOVA or mixed analysis.
I believe study design with more aggressive surgical trauma on jaw would affect systemic polarization, that is mimicking actual condition - High bone turnover and exposure to normal flora. In example, extraction of M1, M2, M3 with alveoloplasty.
Given that the results are not definitive, describing the limitation of the study should be written in discussion section.
Moreover, considering that denosumab doesn't have immunomodulatory effects compared to BP, same or more deteriorating effect for ONJ should be discussed a little bit.
Author Response
'Surgical Trauma' is of concern. Although it doesn't show statistical significance, surgical trauma showed trend of lower CD68, iNOS values. Considering the surgical trauma is main risk factor for ONJ development and also deeply associated with initial inflammation and wound repair, this should not be overlooked. The reviewer recommends additional analysis with adjustment of surgical trauma also with consideration time point such as RM ANOVA or mixed analysis.
Thank you for this comment. We considered indeed to perform a RM ANOVA analysis to compare the different time points in regard to surgical trauma and BP administration. However, the individual groups would be very small and would not allow a sufficient statistical analysis.
I believe study design with more aggressive surgical trauma on jaw would affect systemic polarization, that is mimicking actual condition - High bone turnover and exposure to normal flora. In example, extraction of M1, M2, M3 with alveoloplasty.
Given that the results are not definitive, describing the limitation of the study should be written in discussion section.
This is a relevant point. We added this information in the “limitations of the study” section as you recommended (page 16, line 502ff):
The induced trauma in this study had no significant influence on systemic macrophage polarization. It could be possible that a more aggressive surgical trauma might have stronger effects on the expression of macrophage polarization markers.
Moreover, considering that denosumab doesn't have immunomodulatory effects compared to BP, same or more deteriorating effect for ONJ should be discussed a little bit.
We added this aspect to the discussion (page 15, line 450ff):
However, considering the potential anti-tumoral immune modulating effects of BP indicated by the current study and previous reports, comparisons between anti-cancer effects of BP and Denosumab are needed. This is particularly important because Denosumab can also lead to osteonecrosis of the jaw. Moreover, comparing analyses of Bisphosphonates and Denosumab on polarization and function would be desirable.
Literature:
- Jayaraman P, Alfarano MG, Svider PF, Parikh F, Lu G, Kidwai S, et al. iNOS expression in CD4+ T cells limits Treg induction by repressing TGFbeta1: combined iNOS inhibition and Treg depletion unmask endogenous antitumor immunity. Clinical cancer research : an official journal of the American Association for Cancer Research. 2014;20(24):6439-51.
Round 2
Reviewer 1 Report
The addition of Figure 3 improved substantially the clarity of the paper.
The answers given clarify some points. I believe now the manuscript is suitable for publication.